



# Detecting most effective cleanup locations using network theory to reduce marine plastic debris: A case study in the Galapagos Marine Reserve

Stefanie L. Ypma[1], Quinten Bohte[1], Alexander Forryan[2], Alberto C. Naveira Garabato[2],
Andy Donnelly[3], and Erik van Sebille[1]

[1]Institute for Marine and Atmospheric Research Utrecht, Department of Physics, Utrecht University, Utrecht 3584 CS, Netherlands
[2]Ocean and Earth Science, University of Southampton, National Oceanography Centre, Southampton SO14 3ZH, UK.
[3]Galapagos Conservation Trust, 7-14 Great Dover Street, London, SE1 4YR, UK

**Correspondence:** S. L. Ypma (s.l.ypma@uu.nl)

**Abstract.** The Galapagos Marine Reserve was established in 1986 to ensure protection of the islands' unique biodiversity. Unfortunately, the islands are polluted by marine plastic debris and the island authorities face the challenge to effectively remove plastic from its shorelines due to limited resources. To optimise efforts, we have identified the most effective cleanup locations on the Galapagos Islands using network theory. A network is constructed from a Lagrangian simulation describing the flow of macroplastic between the various islands within the Galapagos Marine Reserve, where the nodes represent locations along the coastline and the edges the likelihood for plastic to travel from one location and beach at another. We have found four network centralities that provide the best coastline ranking to optimise the cleanup effort based on various impact metrics. In particular locations with a high retention rate are favourable for cleanup. The results indicate that using the most effective centrality for finding cleanup locations is a good strategy for heavily polluted regions if the distribution of marine plastic debris on the coastlines is unknown and limited cleanup resources are available.

## 1 Introduction

Marine plastic debris is abundant in the global ocean (Browne et al., 2015; Lebreton et al., 2019) and is likely to increase in the coming decades if waste-management systems across the globe do not improve (Lebreton and Andrady, 2019; Lau et al., 2020). Although the spatial distribution of plastic in the ocean is strongly heterogeneous (Law et al., 2010; Goldstein et al., 2013; Buhl-Mortensen and Buhl-Mortensen, 2017; Kaandorp et al., 2022), both model simulations and observations indicate that land-based mismanaged waste is trapped in coastal zones and that in particular positively buoyant plastic is most likely to be found on or near coastlines (e.g. Lebreton et al., 2019; Onink et al., 2021; Morales-Caselles et al., 2021). This implies that beach cleanups can be a highly effective mitigation measure to minimise the negative impact of plastic pollution on marine coastal ecosystems (e.g. Haarr et al., 2019; Kaandorp et al., 2022). Furthermore, beaches are more accessible and cheaper to clean than the open ocean or the seabed and targeting specifically macroplastic items (>0.5 cm) will prevent the formation on





the shorelines of microplastic which are more difficult to remove (e.g. Andrady, 2011; Kataoka and Hinata, 2015; Ryan et al., 2020; Jones et al., 2021).

As beach cleanups often rely on the availability of professionals and willingness of volunteers, the location and timing of the cleanup activities do not always correspond with the location where cleanups would have most impact (Critchell et al., 2015; Nelms et al., 2017). Kaandorp et al. (2022) explained litter quantities observed along the Dutch North Sea coast by combining a large data set from beach cleanups with hydrodynamic data and virtual particle pathways. Doing so, they were able to give recommendations to optimise cleanup efforts in the Netherlands. Unfortunately, using this data-driven approach to find effective cleanup locations is only possible if a large and consistent monitoring data set is available, which is often not the case (e.g. Canals et al., 2020; Morales-Caselles et al., 2021).

Here, we propose a different approach to develop a cleanup efficacy model when the source and sink distribution of plastic is unknown, using a network that is constructed from a transition matrix describing the flow of macroplastic between different coastlines. In particular, the network analyses used in ecology to study the connectivity of species between spatially separated habitats (Kininmonth et al., 2010; Jönsson and Watson, 2016; O'Malley et al., 2021) bears many parallels with the connectivity of potential macroplastic sources and sinks between various coastlines. The residence time of macroplastic at the shoreline is not infinite and beached macroplastic can resuspend (e.g. Kataoka et al., 2015). Dispersed by the ocean currents, locally littered or beached macroplastic can therefore spread from one coastal site to another, just as e.g. larvae disperse from one habitat to another (Pata and Yñiguez, 2021). Where ecological connectivity is used to determine which regions are in need of conservation to increase the resilience against anthropogenic disturbances (Chamberlain et al., 2021), a network based on coastline connectivity of beached and resuspended macroplastic has the potential to find important cleanup locations on land. If the source distribution of macroplastic is unknown, the connectivity can still, to a first order, indicate which regions are likely to accumulate macroplastic and from which regions macroplastic is likely to spread to other coastlines.

One way to find the most effective clean up location is to implement sinks in the modeled network and compute the amount of macroplastic removed from the system. Searching for the optimal sink locations is often computationally unfeasible as it would require too many iterations to test all possible sink combinations (Sherman and van Sebille, 2016). Instead of using a brute-force method, it is also possible to rank the possible sink locations by importance based on specific network centralities (see for equivalent in conservation efforts of habitats e.g. Watson et al. (2011); Treml et al. (2008); Pata and Yñiguez (2021)). However, depending on the management objective, it is not always straightforward to determine which centrality leads to the desired ranking that has highest impact.

In this paper, we construct a network based on the flow of macroplastic (>0.5 cm) between the coastlines of various islands within the Galapagos Marine Reserve (Figure 1). The marine reserve and the Galapagos Islands are a UNESCO World Heritage Site and host a unique biodiversity (e.g. Denkinger and Vinueza, 2014). The islands face increasing levels of pollution mainly from remote sources (van Sebille et al., 2019; Escobar-Camacho et al., 2021), where over 8 tonnes of plastic are removed from the coastlines of the Galapagos Islands each year (estimate from unpublished coastal clean-up data). Recent observations have identified 27 Galapagos marine vertebrates at high risk by exposure to microplastic contamination (Jones et al., 2021).





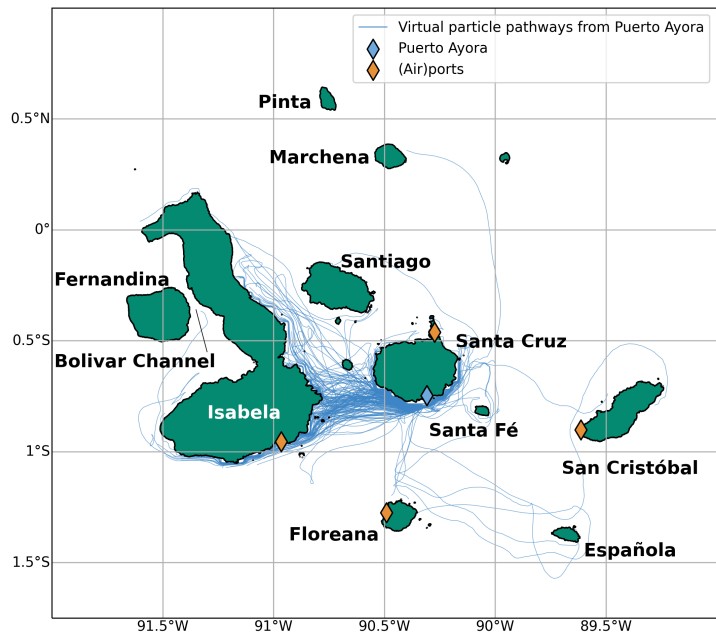

**Figure 1.** Map of the main islands in the Galapagos Marine Reserve. All port locations are indicated by diamond markers and virtual particle pathways (blue lines) show all existing connections via ocean currents using the MITgcm model simulation from Puerto Ayora (blue diamond) to other coastlines.

Although monitoring efforts to measure macroplastic abundance along the Galapagos coastlines are increasing, the macroplastic distribution is still largely unknown due to limited resources. Conservation efforts will therefore strongly benefit from identifying key locations for cleanup. Previous studies have shown that a strong inter-island genetic connectivity exists within the Galapagos Marine Reserve, mainly due to seed dispersal by ocean currents (e.g. Fajardo et al., 2019; Arjona et al., 2020). Thus, it is likely that also macroplastic may have high residence timescales within the marine reserve due to ocean transport

processes between the islands. Therefore, the connectivity between the coastlines of the Galapagos Islands provides an ideal case study to investigate the use of centrality theory to find optimal cleanup locations.

    In this paper we aim to identify which centralities provide the best ranking to optimise cleanup efficacy at coastlines of the Galapagos Islands. The resulting cleanup efficacy-model will provide a valuable first indication of which locations to target and can easily be applied to other regions across the globe.

**2   Methods**

In order to set up the network of macroplastic connectivity between the Galapagos Islands, virtual particles are released from all island coastlines. The pathways of the particles are constructed using a Lagrangian tracking tool (section 2.1) where the



arrival of particles at the coastline is parameterised by a beaching parameterisation (section 2.2). From these pathways, a transition matrix is constructed that provides the likelihood that two coastlines are connected (section 2.3). It is then possible to find the optimum cleanup locations using various coastline rankings based on centrality measures (section 2.4). To test the impact of these rankings, three metrics are proposed in section 2.5 that optimise for maximum removal of particles and minimal distribution of particles on land.

## 2.1 Lagrangian particle simulation

The transition matrix used for the network analysis is constructed from a Lagrangian simulation of macroplastic transport between the Galapagos Islands. In this paper we only consider positively buoyant macroplastic (>0.5 cm) as larger pieces of plastic are the targets of the beach cleanup (see section 1). As such, we used the two-dimensional daily-mean surface velocity fields from the regional MITgcm model described in detail by Forryan et al. (2021). The model has a horizontal resolution of 4 km and is forced at the three open boundaries (north, south and west) by the temperature, salinity and velocity fields of the ORCA0083-N01 1/12° NEMO model[1]. Surface forcing from ERA-Interim at 6-hourly resolution was applied and the simulation years 2008–2012 were used for the particle tracking. The model is able to capture the main surface circulation and variability observed near the Galapagos Islands and is therefore well suited to investigate the MPD connectivity between the islands (Forryan et al., 2021).

Virtual particles were released daily throughout the five years of the model simulation at every coastal grid point of the 10 islands that are resolved by the MITgcm model (see Figure 1). A coastal grid point is defined as the model grid point that shares one or more corners with a land grid point. The particles are advected for 60 days through fourth-order Runga-Kutta integration in the *Parcels* Lagrangian framework (Delandmeter and van Sebille, 2019) with an hourly output frequency. A two month timescale has proven sufficient to connect any two coastal grid points within the marine reserve. In total, 707.400 particle trajectories (1800 days x 393 coastal grid points) were used to construct the transition matrix.

As the timescale to travel from one coastline to another is at most two months, processes that impact the vertical movement of macroplastic, such as fragmentation processes and biofouling are not included as these typically have a much longer timescale (e.g. Andrady, 2011; Song et al., 2017; Gerritse et al., 2020; Kaandorp et al., 2020; Lobelle et al., 2021). Further, we have chosen not to include any artificial Lagrangian diffusion as the mesoscale variability is largely resolved by the flow fields. We also did not take into account effects of wind drag on floating plastic or the role of Stokes drift. Although these effects likely play a role for macroplastic transport in the marine reserve, it is unclear how they should be parameterised as there are only few observations from floats near the Galapagos Islands available for validation (van Sebille et al., 2019). We have therefore chosen to solely use the Eulerian surface flow for constructing the network, but the method can be easily extended.

## 2.2 Beaching parameterisation

To construct the transition matrix we compute the likelihood that a particle entering the ocean from one location will beach at another location (or at the same location). We are only interested in the particle arrival as this would provide information on

---

[1]https://www.nemo-ocean.eu/





whether a cleanup is needed. We therefore consider the beaching processes and not the resuspension process for the construction of the transition matrix in section 2.3.

As the horizontal model resolution is only 4 km, the beaching process is parameterised using the same method as described by Onink et al. (2021): if a particle reaches a coastal grid point, a beaching probability $p_B$ to determine whether the particle is beached is calculated as:

$$p_B = 1 - exp(-dt/\lambda_B),$$

where $dt$ is the integration time step of the Lagrangian simulation (1 hour) and $\lambda_B$ the beaching timescale. Although observations for the validation of this parameterisation are limited, analysis by Pawlowicz et al. (2019) of a set of beached drifters in the Salish Sea showed that this parameterisation provides reasonable estimates for the beaching location. The choice for the beaching timescale however is uncertain and so far, values of $\lambda_B = 1$ - 100 days have been used (e.g. Kaandorp et al., 2020; Onink et al., 2021; Kaandorp et al., 2022). It seems that mainly quantitative analyses such as mass-budgets are sensitive to the choice of the beaching timescale, but that beaching patterns are qualitatively robust against the choice for $\lambda_B$. In this paper we use a range of plausible beaching timescales, $\lambda_B \in [1, 2, 5, 10, 26, 35]$ days, to test the sensitivity of our transition matrix on the choice of $\lambda_B$. We have decided not to test for $\lambda_B > 35$ days as our advection timescale is only 60 days.

## 2.3 The transition matrix and corresponding network

Combining the Lagrangian simulation (section 2.1) and the beaching parameterisation (section 2.2) we construct a transition matrix that provides the likelihood that a particle travels from a source coastal grid point $n_{so}$ (hereinafter a source node) and beaches at a sink coastal grid point $n_{si}$ (hereinafter a sink node, Figure 2a). To ease interpretation, all nodes are grouped per island (from east to west) and ranked in a clockwise direction starting at the northernmost node of each island.

Between Isla Isabela and Isla Fernandina a narrow strait exists (the Bolivar Channel, see Figure 1), which is only one ocean grid point wide in the hydrodynamic model. As the beaching parameterisation can only provide information on beaching within one grid point from land, the channel is chosen as a separate source/sink location.

The transition matrix is then translated to a directed network, where each node represents a coastal grid point and the edge weight $p_{so,si}$ the probability that a particle travels from $n_{so}$ to $n_{si}$. Note that using this method, only connections exist between two nodes if the travel time is below the maximum particle advection time of 60 days. As mentioned before, increasing the advection time does not change the conclusions drawn in this paper. As such, we are not concerned about the time scale, but purely on whether a connection between two nodes exists.

The directed network is constructed using the range of beaching timescales introduced in section 2.2. A larger beaching timescale reduces the likelihood of pathways between the nodes (Figure 2b), but the connectivity pattern itself seems insensitive to changes in $\lambda_B$. Therefore, for the remainder of this paper we present results using $\lambda_B = 5$ days, where 42% of the particles beach.





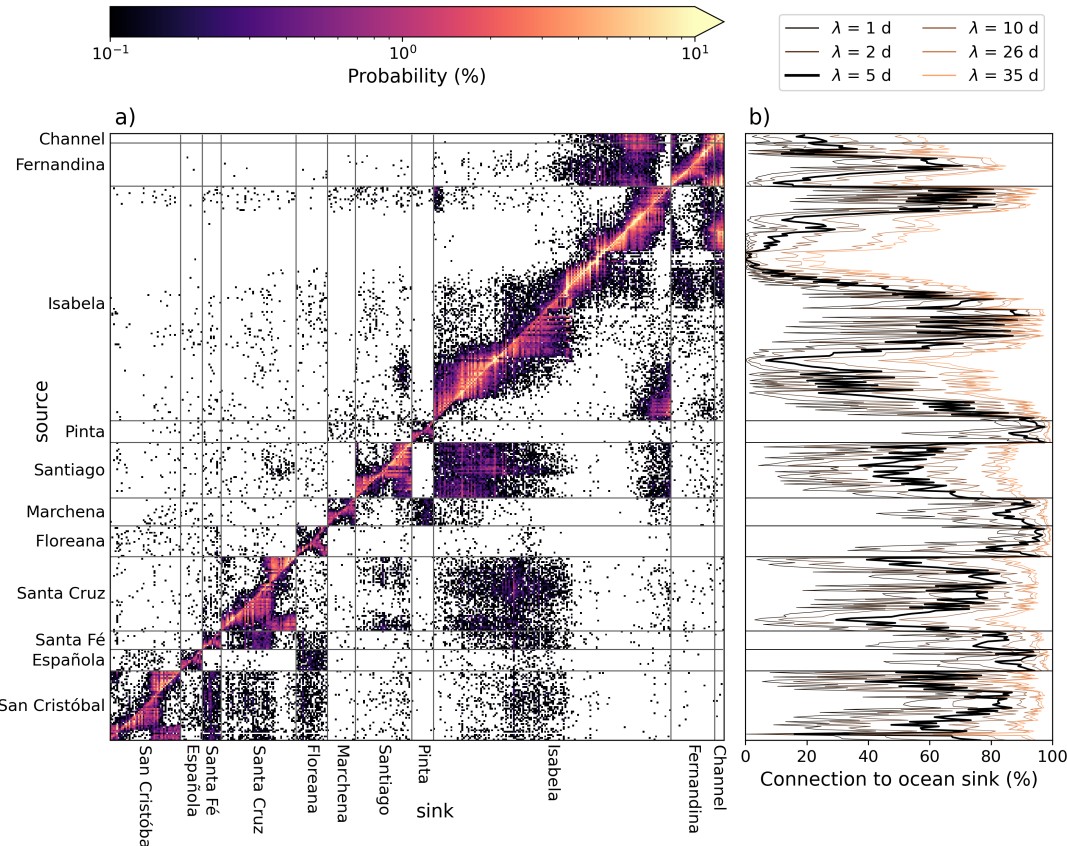

**Figure 2.** The transition matrix (a) giving the probability that a particle starting at a source coastal node (y-axis) arrives at a sink coastal node (x-axis) for $\lambda_B$ = 5 days, and the probability (b) that a particle starting at a source coastal node is not beaching within 60 days and therefore 'lost' to the ocean, including the sensitivity of this loss to changes in the beaching timescale (colored lines) compared to the reference simulation (black line). The different islands are delimited by horizontal and vertical grey lines.

## 2.4 Network centralities to identify effective cleanup locations

To identify which locations should be cleaned first to reach maximum impact, a ranking of all coastline nodes is established based on various network centralities, where nodes with a higher centrality value are cleaned first. Whenever relevant, the
130 NetworkX package version 2.5.1 is used to calculate the centrality from the network constructed in section 2.3 (Hagberg et al., 2008). We chose ten centralities for which the description and motivation is provided below.

– The **Retention Rate (retention)** indicates the percentage of particles for which the sink node is the same as their source node. Nodes with a Retention Rate equal to one have no outgoing edges, but may have incoming edges, and represent regions where particles are very unlikely to leave.





– The **Loss Rate (loss)** is the percentage of particles that are lost to the ocean, i.e. the edge weight of the edge that connects each node to the ocean (Figure 2b). Nodes with a Loss Rate equal to one have no outgoing edges to coastline nodes, but may have incoming edges. These nodes may represent regions that should be cleaned rather fast to avoid loss to the open ocean where removal is more difficult.

– The remote **Beaching Rate (beaching)** is the percentage of particles that have a coastline sink node that is different

from their source node. Prioritising nodes with a high remote beaching rate for cleanup may have a non-local impact by preventing the spread of particles to other locations.

– The **Source-Sink Index (SSI$_{sink}$ and SSI$_{source}$)** is given by

$$SSI = \frac{(P_{in} - P_{out})}{P_{in} + P_{out}},$$

where $P_{in}$ and $P_{out}$ give the total edge weight of all incoming and outgoing edges respectively. Positive SSI values (**SSI$_{sink}$**) indicate that a node is a net sink, whereas negative SSI values (**SSI$_{source}$**) indicate that a node is a net source (Pata and Yñiguez, 2021). Comparing the impact of these two centralities will give insight in whether removal at a net

source or at a net sink is more important.

– We use the **Sink Diversity (SiD)** and **Source Diversity (SoD)** as described by Pata and Yñiguez (2021) to identify nodes that have outgoing edges to many different sinks with a high edge weight (high SiD) or to identify nodes that have strong incoming edges from many different sources (high SoD). The index is computed from a modified Shannon's diversity index (Holstein et al., 2014) using

$$SiD_{so} = -\sum_{i=1}^{si} p_i \ln p_i$$

and

$$SoD_{si} = -\sum_{j=1}^{so} p_j \ln p_j,$$

where $p_i$ and $p_j$ are the weights of each outgoing and incoming edge normalised by the sum of all outgoing and incoming edge weights at each source and sink node.

– The **Pagerank Centrality (PR)** is calculated to quantify the importance of each node proportional to the importance of its neighbours. To calculate this importance, often the eigenvector centrality is used (e.g. Watson et al., 2011; McAdam

and van Sebille, 2018), but pagerank centrality is more appropriate to use in case of directed networks (Newman, 2018) as it can give more importance to incoming edges than to outgoing edges. As it is a priori not clear whether an incoming edge or an outgoing edge is more important for finding effective cleanup locations, the pagerank centrality is calculated from both the original directed network (**PR$_{in}$**) as its transpose (**PR$_{out}$**).

– Lastly, we use the **Betweenness Centrality (betweenness)** that, instead of considering the in- and out-degree of the various nodes, gives more importance to nodes that lie on shortest paths between other nodes. In other words, the





betweenness centrality could help find regions where it is likely that a large amounts of particles pass by. This centrality has been used before to find effective cleanup locations along the shorelines of South Korea by Jeon et al. (2015). However in contrast to their method, we define the shortest path based on the most likely path instead of the shortest distance or travel time as we want to optimise for maximal removal of particles, not for e.g. minimal residence time of particles in the ocean. By simply rewriting the edge weight probability $p_{so,si}$ as

$$w_{so,si} = -log(p_{so,si}),$$

regular shortest path algorithms can be applied to calculate the betweenness centrality, but now based on the most likely path (e.g. Ser-Giacomi et al., 2015; O'Malley et al., 2021).

## 2.5 Impact metrics

As a management objective we define the most effective cleanup locations as regions where cleanups minimise the macroplastic distribution on land and maximise the total amount of macroplastic removed. Furthermore, as there are only limited resources available for cleanup, we need to take into account how much of the coastline can be cleaned, and how often, to reach a specific 160 macroplastic distribution.

The particle distribution on coastlines $\boldsymbol{v_{t^*}}$ at a time $t^*$ can be calculated by multiplying the previous distribution with the transition matrix as

$$\boldsymbol{v}_{t^*} = \boldsymbol{v}_{t^*-1} A_{so,si}.$$

Note that the time $t^*$ does not represent an actual time as the travel time for specific pathways is variable and we did not include any residence time scales for particles on shorelines. It does however provide an indication for how often specific locations should be cleaned until there are no more particles arriving from other locations.

We initialise the system with a homogeneous particle distribution before cleanup starts. Then, using the node rankings from 165 the various centralities, the outgoing edges of the cleanup target nodes are removed. This method bares similarities with the percolation process in network theory used to study clusters (Newman, 2018). However, in this paper we leave the incoming edges intact. After every iteration, particles will accumulate at the target cleanup node until a steady state is reached, which provides a means to quantify the cleanup impact.

To ensure that we perform sufficient iterations to reach steady state, we calculated the change in particle number in the ocean 170 between two iterations. We then define to have reached steady state when this change is smaller than 0.1% of the total number of particles.

Three impact metrics are used to evaluate the efficacy of the different centrality node rankings. The **benefit** metric indicates the difference (in %) between the total number of particles removed and the number of particles removed if there had been zero connectivity between the different nodes. The **Left Behind on Land** metric gives the percentage of particles that are still 175 on land when a steady state is reached. Then finally, the **iteration** metric provides an estimate of how often the calculation for the next particle distribution is performed to reach a steady state.





For a given percentage of coastline that is cleaned, the most effective centrality node ranking would maximise the **benefit** metric (total number of particles removed) and minimise the **Left Behind on Land** and **iteration** metrics (minimal particle distribution on land and only little repeated cleaning needed).

## 3 Results

### 3.1 Source and sink distribution of buoyant macroplastic connectivity

The transition matrix and corresponding network constructed in section 2.3 provide a first indication of the connectivity struc-ture (i.e. the network topology) of the macroplastic flow between the coastlines of the Galapagos Islands (Figure 2a). The network is characterised by short source-sink distances, as the highest probabilities in the transition matrix are found along the

diagonal. Although the network is relatively sparse, the number of strongly connected components in this network is equal to one, which means that there always exists a pathway connecting any two nodes in the network. Furthermore, most connections between nodes are weak (i.e. pathways with low probability) and only a few are strong (i.e. pathways with a high probability). Therefore, the distribution of all pathway probabilities (i.e. the edge weights) is relatively linear in a log-log plot (not shown) indicating that the macroplastic connectivity can be classified as a small-world network (e.g. Watts and Strogatz, 1998; Amaral

et al., 2000; Kininmonth et al., 2010; Watson et al., 2011).

The ocean is the largest sink for particles leaving the islands (Figure 3a) when using a beaching timescale of $\lambda_B = 5$ days (note that the total percentage lost to the ocean decreases for smaller beaching timescales, Figure 2b). The amount lost to the ocean, and therefore also the connectivity to other island coastlines, is however strongly spatially variable. Small islands show a stronger connectivity to the ocean than larger islands (Figure 3a). The larger islands show interesting spatial connectivity

variability. At the islands of San Cristóbal, Santa Cruz and Fernandina, the largest sink to the ocean is found at southern locations, whereas Santiago Island loses most particles to the ocean in the north and the sink distribution of Isabela Island shows a large loss to the ocean both in the north and in the south. This is most likely a result of the relative orientation of the islands with respect to the ocean circulation pattern.

The circulation pattern near the Galapagos Marine Reserve is mainly dominated by the westward South Equatorial Current

in the east, and by upwelling of the Equatorial Undercurrent waters in the west (Chavez and Brusca, 1991; Liu et al., 2014; Forryan et al., 2021). Indeed, the island-to-island connectivity is mainly in a westward direction (Figure 2a) and in general, more particles arrive at islands downstream than upstream (Figure 3b). Also, the source distribution (Figure 3c) reveals that particles arriving at Isabela Island are more likely to come from other islands (>20%) than the particles arriving at San Cristóbal Island (2.0%).

For some of the islands (San Cristóbal, Santiago and Fernandina), more particles beach at the western shorelines than at the eastern shorelines (Figure 3b). This might seem contradictory to previous work that suggests that most macroplastic arrives at the eastern shorelines due to the prevailing ocean current direction (e.g. van Sebille et al., 2019; Jones et al., 2021). However, here, the initial distribution is homogeneous and therefore more representative of potential local sources of macroplastic instead





**Figure 3.** Source (a) and sink (c) distribution of macroplastic connectivity between the Galapagos Islands, specifying the relative connectivity to/from another island (green), to/from the same island (yellow), to/from the same location (brown) and to the ocean (blue). The total percentage of particles arriving at each location (node) is shown in panel b.





of remote (i.e. sources from outside the marine reserve). Apparently, without an influx of remotely sourced macroplastic, the
local dynamics leads to accumulation at the western coastlines at these islands.

San Cristóbal Island and Santa Cruz Island display a very similar sink distribution pattern (Figure 3a). The particles seem to
travel from one node to another in a clockwise direction at the eastern flank and a counter-clockwise direction at the western
flank of the islands. They follow the shoreline, until they reach a 'take-off' node, i.e. the location where the particles are
most likely to leave the island. The negative correlation between the likelihood to beach at another island and the likelihood
to stay close to the source is also visible in Figure 3a. This indicates that the connectivity between the nodes that belong to
the same island is fundamentally different from the island-to-island connectivity. This is most likely a result of predominantly
low velocities close to the shorelines in combination with the beaching parameterisation that is applied only close to shore.
The 'take-off' locations might be interesting targets for cleanup strategies as they seem to be accumulation points for particles
originating from the same island, and at the same time display a strong out-flux of particles to the ocean and to other islands.

**3.2 Impact of cleanup based on the coastline node rankings**

The visual representation of the source and sink distribution of macroplastic connectivity already provides some insight into
potential cleanup locations. However, a more quantitative analysis can be performed by iteratively removing the outgoing edges
from specific nodes (i.e. the cleanup target locations) and assessing the impact as described in section 2.5. Which outgoing
edge to remove first is determined by the ranking of the coastline nodes given by various centralities (see section 2.4).

The impact metrics reveal that the cleanup efficacy is strongly dependent on the order in which the coastline nodes are
cleaned. As expected, but important to note, is that due to the connections between the different potential cleanup locations,
there is always a positive benefit from cleaning only a fraction of the coastline (Figure 4a). For example, cleaning 30% of the
coastline based on the $SSI_{sink}$ ranking can lead to a reduction of >50% of the total number of particles present in the system
(red line in Figure 4a). This means that the cleanup effort can indeed be optimised using limited resources.

Which centrality to use for the most efficient cleanup strategy however depends on how much of the coastline can be cleaned.
Overall, the $SSI_{sink}$ seems to provide the highest benefit (red line in Figure 4a). However, for low cleanup effort (<10% of the
coastline), the betweenness centrality (grey line), the retention rate (blue line) and the pagerank centrality favouring incoming
edges ($PR_{in}$, yellow line) lead to similar benefit rates. For high cleanup effort (>90% of the coastline) the loss rate seems
to outperform the $SSI_{sink}$ regarding the benefit rate. Interestingly, the particles that would *not* be removed are still on land
when using a cleanup strategy based on the loss rate, instead of being lost to the ocean if the $SSI_{sink}$ ranking is used for the
cleanup strategy (Figure 4b). This demonstrates that depending on the management objective and the cleanup capacity different
centralities might be of importance.

Due to the remoteness of the Galapagos Islands, we would be mainly interested in a cleanup with limited effort and we will
therefore focus on efforts that would be able to clean up to 10% of the coastline. Furthermore, as described in section 2.5, we
do not only want to optimise our cleaning efforts to reduce the overall amount of macroplastic, but also minimise the amount
of macroplastic left on land and the number of iterations needed to reach steady state.







**Figure 4.** A comparison of the impact metrics described in section 2.5 as a function of the fraction of coastline nodes cleaned for all centrality rankings (colored lines). The benefit metric (a) measures the difference (in %) between the total number of particles removed and the number of particles removed if there would have been zero connectivity between the different nodes. The Left Behind on Land metric (b) indicates how many particles are still on land after steady state is reached. The Iterations metric (c) shows how many iterations where needed to reach steady state and provides an indication for how often one should clean.





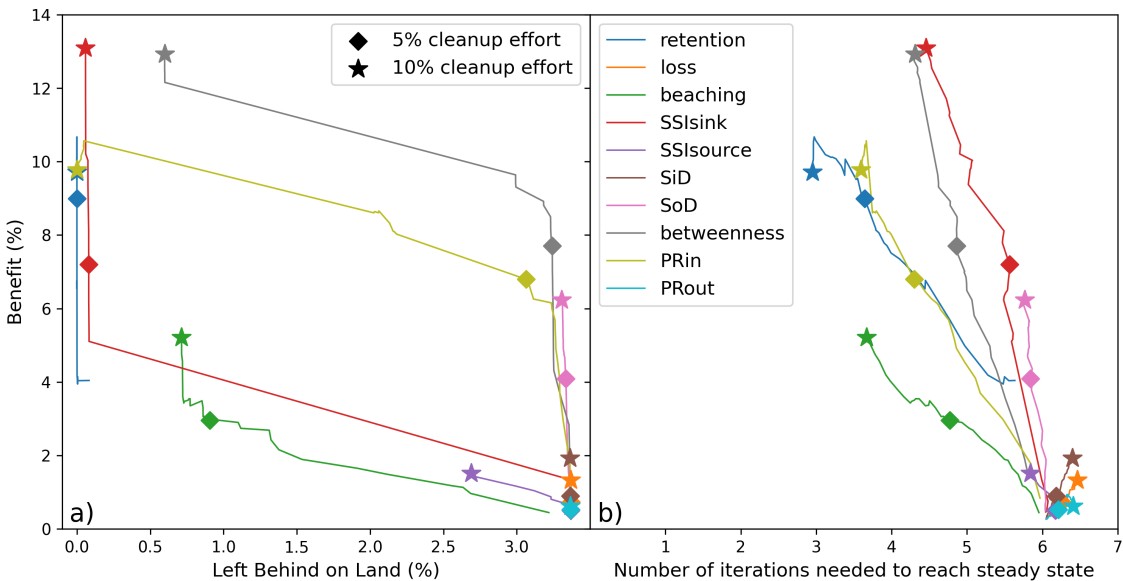

**Figure 5.** The benefit of all centrality node rankings as a function of (a) how much of the initial plastic distribution is left on land and (b) how often the matrix multiplication needs to be performed to reach steady state. The impact metrics are calculated for when 5% (diamond marker) and 10% (star marker) of the coastline would be cleaned.

Using these objectives, the retention rate seems to provide the most effective cleanup strategy for small (5%) cleanup effort (blue diamond marker in Figure 5). Targeting regions with a high retention rate would lead to cleanest coastlines, with the least amount of repeated cleaning and with the highest benefit. This would indicate that the dynamics that keeps macroplastic at the
same location is more important for effective cleanup efforts than the arrival of macroplastic from other locations.

For slightly higher cleanup effort (10%) we again find that the node rankings based on the betweenness (grey marker), the $SSI_{sink}$ (red marker) and the $PR_{in}$ (yellow marker) centralities become important. Apart from the betweenness centrality, these node rankings, together with the retention rate, have in common that they give more importance to incoming edges than to outgoing edges. This is an interesting finding as this implies it might be more effective to clean at a sink than at a source for
the overall macroplastic removal.

### 3.3 Sensitivity to initial macroplastic distribution

So far, we have assumed a uniform initial distribution of particles to evaluate the impact of the various nodes rankings. As mentioned in the introduction (section 1), we are interested in finding a cleanup strategy that can be used when the distribution of macroplastic is unknown. Therefore, we also test the impact of the node rankings against a random distribution of particles





across the coastlines nodes. With a random distribution we mean that every coastal node is initialised with a random particle weight between 0 and 1.

Logically, the potential total particle 'mass' removed is sensitive to how much of the coastline is initially polluted. However, we note that the relative efficacy of the node rankings based on the various centralities stays the same. For example, if 10% of the coastline could be cleaned, targeting those coastline nodes with a high $SSI_{sink}$ centrality or betweenness centrality would

most likely lead to a higher number of particles removed than targeting nodes based on other centralities, regardless of the initial distribution (not shown). Theoretically, this would even mean that by measuring the amount of waste cleaned at these locations, one could infer how much pollution is present in total. Note however, that using this method we do not consider any new sources of macroplastic during the cleanup event, neither from land nor from remote sources. The resuspension time scale for macroplastic to return to the ocean is also not included in this study. Therefore, for long resuspension time scales the total

number of particles may change before a steady state can be reached.

By using a random initial particle distribution, one can also compare the cleanup impact to cleanup efforts that would target the most polluted coastlines (i.e. when the initial distribution was known). Figure 6 provides an example of this comparison when the cleanup effort is 10% and the $SSI_{sink}$ centrality is used to identify cleanup locations. The impact is calculated by determining the total removed particle 'mass' when steady state is reached divided by the total particle 'mass' at initialisation.

The distribution is initialised randomly, where a fraction of the coastline is clean. Then the total particle mass removed is calculated, both when targeting nodes with the highest particle weight and when targeting nodes with a high $SSI_{sink}$ centrality. The advantage of knowing the initial distribution is then given by the difference of the two removal estimates (Figure 6). If the advantage is equal to zero, targeting coastline nodes with highest particle weight has a similar impact as targeting nodes with a high $SSI_{sink}$ centrality.

As expected, targeting nodes with the highest particle weight will almost always have an advantage to using the node ranking based on the centralities. However, there is one important take away from this comparison. The lower the fraction of clean coastline nodes, the less important it is to know the exact particle distribution. Even when targeting locations with a high particle weight, the total removed particle mass is limited by the cleanup effort and the difference with targeting locations with a high centrality ranking is minimal.

## 3.4 Optimal cleanup locations

Finally, the 10% highest ranked coastline nodes given by the various centralities are shown in Figure 7. The coastline locations indicated by the first three centralities (the Retention Rate, the Loss Rate and the Beaching Rate, Figure 7a-c) could have already been inferred from the sink distribution shown in Figure 3a. Indeed, the smallest islands lose most particles to the ocean (Figure 7b) and Isabela Island retains most particles (Figure 7a) and has the highest likelihood of a particle sink on land

(Figure 7c). The impact of the westward South Equatorial Current is most notable for coastline nodes with a high Source and Sink Diversity (Figures 7f and 7g) and a high Pagerank centrality (Figures 7i and 7j).

Interestingly, the mean westward flow does not lead to net sinks in downstream regions or net sources in upstream regions (Figures 7d and 7e). However, these net sinks and sources do seem to be related to the orientation of the islands with respect

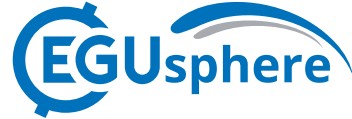

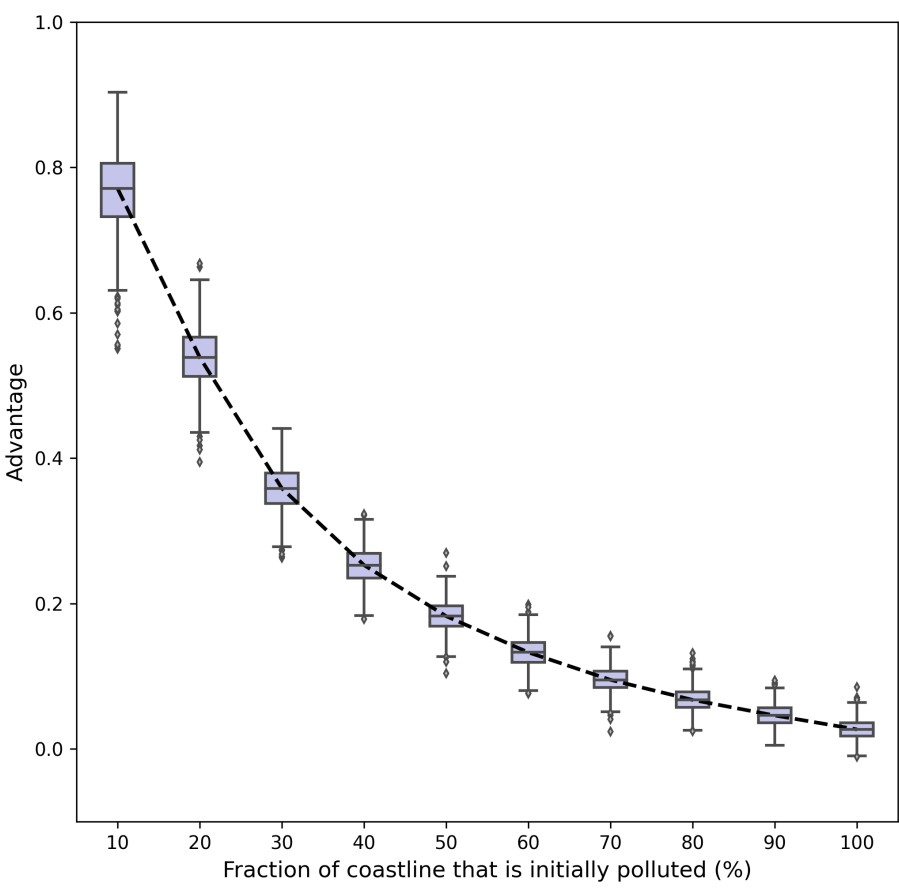

**Figure 6.** The difference between the total removed particle mass if the initial distribution of particles is known and the total removed particle mass when using the $\mathrm{SSI}_{sink}$ centrality. The difference is plotted as a function of how clean the coastline is initially (in %). For this calculation, a cleanup effort of 10% is applied and each calculation is repeated 1000 times with randomly distributed particle weight. Outliers are shown with diamond markers.





**Figure 7.** The 10% highest ranked coastline nodes by the centralities discussed in section 2.4. The coastline locations determined by the Retention Rate (a), the $SSI_{sink}$ centrality (d), the betweenness centrality (h) and the $PR_{in}$ centrality (i) are recommended cleanup locations for the Galapagos Islands management that potentially have high impact (section 3.2).



to the mean flow. The $SSI_{sink}$ centrality reveals that the southern region of San Cristóbal Island and Santa Cruz Island and the
northern and southern region of Isabela Island are strong net sinks (Figure 7d). These locations correspond with the 'take-off'
locations discussed in section 3.1. As the $SSI_{sink}$ centrality ranking has a high impact (section 3.2), these 'take-off' locations
are indeed effective cleanup targets as hypothesised.

From the impact assessment of section 3.2 we can conclude that locations with a high Retention Rate (Figure 7a), $SSI_{sink}$
(Figure 7d), betweenness centrality (Figure 7h) and $PR_{in}$ centrality (Figure 7i) are the most effective cleanup locations. Where
the locations based on the Retention Rate and the $PR_{in}$ centrality are confined to specific areas on Isabela Island, the locations
based on the betweenness centrality and $SSI_{sink}$ are more scattered across the islands. The solution to finding the most effective
cleanup locations on the Galapagos Islands can therefore be optimised based on additional management objectives, such as
how easily accessible effective cleanup locations are and objectives related to the local environmental impact of macroplastic.

## 4 Discussion and Conclusions

In this paper we have identified which centralities are useful to optimise beach cleanup efficacy for macroplastic removal
based on macroplastic flow connectivity between the coastlines of the Galapagos Islands. We have identified four centralities,
the $SSI_{sink}$ centrality, the Retention Rate, the $PR_{in}$ centrality and the betweenness centrality (see for definition section 2.4)
that perform well. We have shown that using the most effective centrality for finding cleanup locations is a good strategy if
the distribution of macroplastic on coastlines is unknown, in particular if only limited cleanup resources are available and the
coastlines are strongly polluted. This means that the cleanup management can focus on repeated cleaning at a few key locations
instead of needing to travel to different locations each time.

Due to the increasing tourism on the Galapagos Islands (Escobar-Camacho et al., 2021) and the waste management chal-
lenges the islands face (e.g. Mestanza-Ramón et al., 2020; Wang et al., 2021), it is important to not only unravel the waste
streams on the islands and the mainland, but also understand the waste streams in the ocean if the waste is not properly man-
aged. The effective cleanup locations presented in Figures 7a, 7d, 7h and 7i therefore do not only indicate regions where a
cleanup can be effective, they also represent locations where good local waste management is very important.

There are however several limitations to the methodology presented in this paper. As we did not include any resuspension
time scales, it is not possible to give recommendations on how often or how long specific locations should be cleaned for.
Observations of typical resuspension time scales are rare (e.g. Kataoka et al., 2015; Hinata et al., 2017) and, just as for the
beaching time scale, the resuspension time scale depends on the geomorphology of the coastline (e.g. Brennan et al., 2018;
Weideman et al., 2020).

Furthermore, we did not include the effect of wind and waves on the movement of macroplastic through the marine reserve
and neglected the impact of tides. These limitations could to some extend be removed if observational data sets are improved
for both flow patterns of macroplastic through the reserve and estimates of macroplastic retention rate on coastlines of the
Galapagos Islands. These would enable us to validate and adjust the connectivity matrix constructed in this paper. Further,





estimates of resuspension time scales could be incorporated in the iteration process to calculate the impact of the centralities of interest (section 2.5).

Finally, we want to stress that the network centralities are only useful if one can assess their impact. This paper presents a means to do so, but also shows that the impact itself, and therefore the most succesful centralities, depends on the management
objectives posed. Next steps would be to apply the connectivity method described in this paper to other regions across the globe. It would be interesting to see whether other island nations show a similar network structure and whether the same centralities lead to effective cleanup locations. This would mean that the optimal centrality measure is independent of local ocean dynamics and could potentially be used for cleanup strategies globally to reduce marine plastic debris.

*Code and data availability.* Code used to conduct the experiment and to create all of the figures is available at
https://github.com/OceanParcels/Galapagos_connectivity, the particle data and MITgcm simulated velocity fields will be made available through https://public.yoda.uu.nl/.

*Author contributions.* S.Y. and Q.B. designed and conducted the study, creation of the MITgcm velocity data was done by A.F. and A.C.N.G.. All authors contributed to the discussion of the analysis and the final manuscript.

*Competing interests.* No competing interests are present

*Acknowledgements.* Funding was provided to S.Y. by Galapagos Conservation Trust and Evolution Education Trust, Pathways to Sustainability and the K.F. Hein Fonds. A.F. and A.C.N.G. acknowledge the support of the Royal Society (CHL/R1/180428).



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
