# Peer review of "Detecting most effective cleanup locations using network theory to reduce marine plastic debris: A case study in the Galapagos Marine Reserve"

_EGUsphere, 2022_

## Author Comment (AC1)

**Reply Comment #1 – Noam Vogt-Vincent**

*Firstly, I think this is a great manuscript and I enjoyed reading it – a novel idea with potentially very useful results for those involved in marine debris management efforts on Galapagos (and other remote islands if this methodology were implemented elsewhere). The effectiveness of the clean-up strategies proposed in this manuscript will depend on the veracity of the assumptions a significant proportion of debris undergoes resuspension, but this is clearly stated in the conclusion. So overall, I think this manuscript will be valuable for island managers, and researchers working in this field. I do have some technical questions/requests for clarification though.*

We would like to thank Noam Vogt-Vincent for going through our manuscript in so much detail. We in particular liked the suggestion made for improving our sensitivity study to initial distributions using a different type of noise: these new results are a nice addition to our manuscript. Our response to each individual comment can be found below.

1. *I wonder whether you carried out a sensitivity analysis to test whether you released sufficient particles? With such high resolution hydrodynamical data forcing your particle-tracking, significant dispersion can occur over a 60-day integration time with an original separation of <4km (e.g. the off-diagonal cells in the transition matrix are 'noisy', and this is probably why). This in itself is not an issue since practitioners will probably not be using your transition matrix, but it would be nice to know how robust, say, Figure 7 is to particle number. I'm guessing that you were limited to 700k particles due to storage (since you were saving particle positions with a very high output frequency) but, if it is tractable, a quick sensitivity test might provide some assurance.*

Using only half the number of particles, we arrive at the same conclusions as presented in the manuscript, indicating that the number of particles used leads to statistically sound results. We have the hypothesis that the 'noisy' off-diagonal cells are a result of the beaching parameterization used, not the limited number of particles. To decrease this 'noisiness', one could decide to produce a multitude of transition matrices that are every time slightly different due to the beaching parameterization and then take a mean. It is however questionable whether this is the right or realistic thing to do, as improving the underlying macroplastic flow model (e.g. improved parameterizations as highlighted by Moulton et al., 2023, taking into account more relevant (atmospheric) processes) will probably lead to more applicable results.

[Figure]

*Figure 1 The transition matrix (left) in the original manuscript and (right) using only half the number of particles. We repeated all analysis with the transition matrix on the right hand-side and all results and conclusions do not significantly change.*

2. *I'm struggling to completely follow section 2.5. If I understand correctly, the equation on line 162 models the distribution of debris on coastlines in the limit of 100% resuspension. You simulate clean-up as removing the outgoing nodes of a clean-up target cell. You then say that "particles will accumulate at the target cleanup node", but how can particles accumulate if you're constantly removing them?*

We agree that the wording we used is confusing and we've rephrased the sentence in the revised manuscript. Instead of 'accumulate at the target cleanup node' we changed the text to say: 'After every iteration, *the incoming particles at the target cleanup node are removed from the system* until a steady state is reached, which provides a means to quantify the cleanup impact.'

3. *It's also not clear to me why you based your definition of steady state on the number of particles in the ocean – does the vector **v**ₜ not reach steady state?*

There are indeed three parameters that could be used to find steady state; the number of particles on land ( $\sum vt$ ), in the ocean and the total number of particles removed. The procedure used to define steady state will give different results depending on which of these parameters is used, in particular when a larger fraction of the coastline is cleaned (see Fig. 2 below). Although the number of iterations needed for reaching steady state change, the main conclusions derived from this figure do not change. We do agree that using the total number of particles on land (**vt**) to define steady state is more intuitive and have changed Fig. 4 and 5 accordingly in the manuscript.

[Figure]

*Figure 2 Steady state analysis.*

4. *I'm also finding the "benefit" metric quite difficult to follow. "Zero connectivity between different nodes", to me, implies that 100% of resuspended debris enters the ocean, but I don't think this is what you meant?*

Yes, this is exactly what we mean. If there would have been zero connectivity, this would imply that all resuspended plastic is lost to the open ocean and does not beach at another location within the marine reserve. Therefore, when one would clean e.g. 30% of the coastline, you would also remove 30% of the pollution at the first iteration (assuming again that the pollution is uniformly distributed along the coastlines). Every iteration following you will clean 0%, as there is no 'new' plastic arriving from other locations. When there is connectivity between regions, you will not only be cleaning the 'local' pollution but also the pollution that is arriving from other locations. We have defined the 'benefit' metric to quantify this additional cleaning impact one can have due to the connectivity between different locations. We've adjusted the explanation in the manuscript to read:

'The benefit metric indicates the difference (in %) between the total number of particles removed and the number of particles removed *if all non-removed particles were directly lost to the ocean after the first iteration.*'

5. *I'm not 100% convinced by your sensitivity tests to the initial macroplastic distribution (3.3). You've tested how robust your method is to uncertainty in the initial distribution by using a completely random initial distribution, i.e. assuming that the mass of plastic on beaches is completely decorrelated across length scales L > 4km. Is this realistic? Your result that the efficacy of node rankings remains the same with a random initial distribution is not surprising to me, since mesoscale ocean structures are much larger than 4km so will on average still see a 'uniform' distribution of resuspended debris. But given that van Sebille et al. (2019) showed that most debris incident to Galapagos arrives from the East, and that there could be large-scale effects from wind shadows, wake eddies, etc., I'd have expected that there would be some large-scale structure in the distribution of debris. I wonder if a more realistic way to model an uncertain initial macroplastic distribution might be by generating perlin noise with a wavelength larger than 4km (e.g. maybe the length scale of an island).*

This is a good suggestion and we've repeated the analysis with a random initial distribution using different correlation length scales within the range of observed spatial scales of marine debris distributions in the ocean (order of 1-10 km, e.g. Kaandorp et al., 2020). We've decided to add the results as a second panel to our previous random initial distribution test, as both sensitivity studies provide different insight (see Fig. 3 below). The former sensitivity test, based on a completely random initial distribution, shows that with an increased fraction of coastline being polluted, using centrality rankings for cleanup efforts becomes more promising. The new sensitivity test, where the initial distribution is correlated across a spatial scale larger than the grid size of the nodes, shows that using the centrality ranking for cleanup efforts is almost always more successful than knowing where most pollution is located. Important to note here is that this is true as long as the cleanup resources are limited (<10% cleanup effort).

[Figure]

*Figure 3 The difference between the total removed particle mass when using the SSIsink centrality and the total removed particle mass if the initial distribution of particles is known. The difference is plotted as (a) a function of how clean the coastline is initially (in %) using a purely random distributed particle weight and (b) a function of the correlation spatial scale used to initialize the random initial distributed particle weight. For both calculations, a cleanup effort of 10% is applied and each calculation is repeated 500 times. Outliers are shown with diamond markers.*

6. *This is not a criticism, but I wonder to what extent using a time-mean transition matrix affects your results. For instance, if we have sites A, B and C each releasing 1 unit of debris per time-step with probability P(A→B)=1/6 and P(B→C)=1/6, the probability of transition P(A→C) is 1/36 (over two timesteps). After 12 time-steps, 1/3 units would have been transported from A→C. But if these transition probabilities were time-varying and turned out to be P(A→B)= P(B→C)=1 during time-steps 1-2 and 0 otherwise, 1 unit would have been transported A→C – 3x more than the time-mean case, even though P(A→B)= P(B→C)=1/6 in both cases when averaged over 12 time-steps. This is obviously an artificially bad case and I completely accept that this goes beyond the scope of your study, but I didn't see a mention of this in your discussion of limitations so I was wondering whether you thought this was a limitation (or if there is evidence that this time variability in the transition matrix is not important).*

This is also a valid point. There is a strong seasonal dependence in the flow fields within the marine reserve, so we would expect the results to change (which locations to target) when using for example monthly transition matrices. However, as soon as you start using these you will 're-introduce' the time dimension, which means you also have to start making choices related to resuspension time scales and the frequency of cleanups. We agree that this is outside the scope of this paper, but it is definitely an interesting next step to consider in addition to investigating the implicit connectivity method raised by reviewer #2 and have added these suggestions to the discussion and outlook section of our manuscript.

**References**

Kaandorp, M. L. A., Dijkstra, H. A., & van Sebille, E. (2020). Closing the Mediterranean Marine Floating Plastic Mass Budget: Inverse Modeling of Sources and Sinks. *Environmental Science & Technology*, *54*(19), 11980–11989. https://doi.org/10.1021/acs.est.0c01984

Moulton, M., Suanda, S. H., Garwood, J. C., Kumar, N., Fewings, M. R., & Pringle, J. M. (2023). Exchange of Plankton, Pollutants, and Particles Across the Nearshore Region. *Annual Review of Marine Science*, *15*(1), null. https://doi.org/10.1146/annurev-marine-032122-115057

---

## Author Comment (AC2)

**Reply Reviewer #1**

*The present work proposes a new methodology for improving beach clean-up, to reduce marine plastic debris. The case study is dedicated to the Galapagos islands and the methodology is using a network to select the optimum criteria for the beach clean- up purposes.*

*The manuscript is of very good quality, well organized with no specific structure problem or methodological problem. Although quite dense for the part explaining the criteria, it is possible for non-specialist to understand the goal of the selected criteria.*

*In summary, the overall quality of this work is up to the expected standards and to my point of view can be published as it is (after some typo and minor possible mistakes). I emphasize the fact that such decision (to my experience of reviewer) is very rare. I therefore thank the authors for having so well prepared their work before submission.*

We thank the reviewer for the positive feedback and valuable comments. Part of our future research is to further improve the macroplastic transport simulations and to incorporate the role of atmospheric conditions and we appreciate the reviewer's suggestions related to these concepts. Our response to each individual comment can be found below.

*However, in order to improve the manuscript, or give ideas for new works, I wish to share some questions or comments:*

1. *a very basic but tricky one to start with: could the authors imagine what the results could be with an irregular grid with a strong refinement close to the coast, or with a high-resolution regular grid of, let say, 300 m in resolution on the horizontal? I reckon that this point could be discussed using some literature. My main concern is that, the present study does not prove that the results are not strongly dependent on the OGCM resolution used. If it were, the overall methodology might provide different scores in terms of criteria?*

Indeed, a tricky question, which is also to some extend raised by reviewer #2 comment (4). A higher near-shore spatial resolution will likely impact the local macroplastic transport, but will have little impact on the transition matrix as the eventual 'beaching' would still be parameterized. E.g., the most probable connections might shift by a few nodes, but the overall structure of the transition matrix, and therefore the most effective centrality rankings will likely not change. The near-shore area is a region where additional coastal processes such as rip currents, swell, and wind shadow zones of islands become important, that, regardless of the spatial resolution, are often not incorporated (Moulton et al., 2023).  We think that it is the inclusion of the latter processes that might impact the scores from the various criteria used.

Regardless, the main aim of the paper is to introduce a network methodology that can assess the impact of cleanup strategies when a transition matrix is available. As mentioned in the manuscript, the 'method can be easily extended' once the macroplastic transport in the near-shore is more realistically represented in the transition matrix. We did make some changes to the introduction, method and discussion section to make our aim of our work and the applicability of the results clearer and would like to refer the reviewer to the track-changed revised manuscript.

Moulton, M., Suanda, S. H., Garwood, J. C., Kumar, N., Fewings, M. R., & Pringle, J. M. (2023). Exchange of Plankton, Pollutants, and Particles Across the Nearshore Region. *Annual Review of Marine Science*, *15*(1), null. https://doi.org/10.1146/annurev-marine-032122-115057

2. *the beaching probability: despite the fact that this has been already published, I am still very skeptical about the tuning of this probability. I am pretty sure that replacing this rather "adamant" formula by a probability calculated on the average wind direction versus the coastline layout in the area would be more realistic...Of course, I understand that there has to be a beaching time-scale of some sort to compensate the lack of grid resolution, but just days, regardless of general weather conditions in the area is strange to me. This is of course just my point of view, but this is what we can see when deploying drifters in general. Of course, this is complicated as the results can be different when looking at different scales, from large to very local scale (at the scale of a beach for example), but for a statistical approach, using regional winds might be ok.*

We thank the reviewer for this suggestion. Part of our future work will focus on the insights from a recent drifter field campaign in the marine reserve to make the beaching parameterization more realistic and we plan to indeed investigate the correlation between local atmospheric conditions, in addition to wave and tidal conditions, for drifter beaching events.

3. *A remark concerning the overall network method: according to some literature, connectivity and oceanic distance can be calculated out of the Lagrangian model "quite" simply, I mean without using a complex machinery. Therefore, it could have been interesting to show some comparisons with some basic diagnostics (overall stranding rate / source identification / oceanographic distance), to better prove the real added value of this complex work presented here. In other words, one might think that rather simple diagnostics could deliver nearly the same message or present similar general results, with a computational cost or calculation cost quite reduced. I understand that the authors might reply that their diagnostics are way refined and accurate, which is probably true, but the fact remains that, in the end, when it comes to mobilizing a cleaning team to go on site, maybe simple diagnostics can already deliver the necessary information for the management? (ok, I am teasing a bit here, but never underestimate the robustness of simplicity!)*

We agree that simple approaches are favorable and can potentially increase applicability. That is why we included diagnostics like the 'retention rate' and 'loss rate', which can be directly deduced from the transition matrix and are easy to understand. As suggested by reviewer #2, we slightly changed the description of these centralities to improve clarity. The 'overall stranding rate' suggested by the reviewer is very similar to the Source-Sink Index proposed in our manuscript.

The highest computational cost is by far the Lagrangian simulation itself, not the calculation of the diagnostics. So, we don't think that using the proposed 'simple' diagnostics will reduce the calculation costs. Furthermore, it is to us unclear how the oceanographic distance can be used as a diagnostic for the cleanup. As we focus on the time-mean system, it makes more sense to use the probability to travel between two nodes instead of the distance (which is time related). Either way, this would still require a Lagrangian simulation and does not provide in our opinion a 'simpler' or easier to interpret diagnostic.

4. *Concerning the seeding of the particles, the random seeding is one interesting case, but when targeting real cases, I am quite surprised that the authors have not tried to perform test cases for which the seeding was increased after heavy rains. If so, they would have put themselves in the position of delivering a real connectivity between islands with identified real sources, i.e. the beaches receiving inland waste through rivers or waste management pipes just after the rain events?*

This is a good suggestion, but not applicable to the Galapagos Islands as the main source of macroplastic arriving at the islands is remote (Sebille et al., 2019, Escobar-Camacho et al., 2021). The 'real sources' therefore can only be identified by either high resolution spatial and temporal observations of marine debris abundance along the coastlines (which is currently lacking), or by long-term simulating macroplastic transport pathways form the mainland and fishing activity towards the islands. The latter is tricky, as there are many unknowns in the source abundance and variability. Therefore, our overall aim of our studies is to combine the presented methodology with predicting the episodic arrival of high-concentration macroplastic patches. Heavy rains might indeed be important for the variability in e.g. river outflow from the mainland and would be interesting to incorporate in the predictive system.

Sebille, E. van, Delandmeter, P., Schofield, J., Hardesty, B. D., Jones, J., & Donnelly, A. (2019). Basin-scale sources and pathways of microplastic that ends up in the Galápagos Archipelago. *Ocean Science*, *15*(5), 1341–1349. https://doi.org/10.5194/os-15-1341-2019

Escobar-Camacho, D., Rosero, P., Castrejón, M., Mena, C. F., & Cuesta, F. (2021). Oceanic islands and climate: Using a multi-criteria model of drivers of change to select key conservation areas in Galapagos. *Regional Environmental Change*, *21*(2), 47. https://doi.org/10.1007/s10113-021-01768-0

5. *In addition, I think that, as the final goal is to deliver advice for where and when to go for beach cleaning, one missing part can be the part looking at the correlation between the weather conditions and the stranding or accumulation close to the shore. This is a temporal advice that can be more efficient than the cleaning frequency advice that can be deduced from the present work. Indeed, as mentioned in their introduction, waste distribution is highly heterogeneous, and one cause is the high variability of regional to local weather conditions. Therefore, one could think that the general cleaning plan could be quite different if the weather conditions variations were selected as one of the key parameters.*

We would like to refer the reviewer to our reply to comment (2). We agree that a better understanding of the role of local atmospheric conditions might aid to an even more effective cleanup strategy then repeated cleanup activities and have incorporated this suggestion in the discussion section of our manuscript:

*'In addition, local atmospheric conditions can play an important role for both beaching and resuspension of macroplastic. The presented methodology to assess the removal impact is based on an explicit connectivity network where the edge weights are constant between iterations. As not only the resuspension and beaching timescales are likely to vary in time, but also the probability of pathways between the various nodes, it would be interesting to extend the impact assessment methodology to allow for a time-varying connectivity network.'*

---

## Author Comment (AC3)

**Reply Reviewer #2**

*This manuscript presents an interesting statistical analysis of the connectivity patterns in the Galapagos to infer the coastal locations with a highest probability of accumulating marine macroplastics. In particular, the authors combine concepts from the graph theory, I.e, centrality derived metrics, and a hydrodynamical model to provide a map with the coastlines of the Galapagos Marine Reserve classified according to the optimization in removing marine litter.*

*Some of the findings are: provide a methodology to cleanup strategy that can be applied even when there are not data about the distribution of macroplastics; among the centralities metrics, the retention rate provide the most useful information to localize regions for cleanup; it is more effective in terms of removing macroplastic to clean at sink points (coastlines with large positive Source-Sink index values: high SSI_sink values) than at source locations.*

*In general, the authors present a interesting work aimed at improving the Lagrangian identification of coastal locations with high probability to find macroplastic advected by the ocean flow. The strategy adopted by the authors, i.e. the use of outputs of a hydrodynamical model and the methodology employed through the network theory is technically sound, turning out to be appropriate for the scope of this work. This is a good piece of work which could be of interest to some OS readers. However there are some weakness in the manuscript an a revision has to be addressed before publication. An improvement of the methodology description is strongly recommended before a new submission. Some aspects related to the methodology should be better supported and discussed more appropriately in the context of previous literature. The structure and organization of the Introduction section lack in clarity, where sentences are repeated. I think the paper could be considered for publication after major revision. The main issues that need to be clarified by the authors are listed bellow.*

We thank the reviewer for the positive and useful feedback and our response to the individual comments can be found below. We have incorporated the specific comments regarding the clarity of the introduction section.

1. *I am not sure if the size of macroplastics (>0.5 cm) allows the macroplastic particles to be considered as Lagrangian particles? I think that to compute Lagrangian particle trajectories one needs to assume that the particle has to instantaneously follow or to be totally constrained to the motion of the fluid flow. Even for particles with finite size, the Maxey-Riley approximation to resolve the equation of motion for inertial particles, assumes that particles are very small (small microplastics?).*

The reviewer is correct that the construction of our transition matrix is based on virtual particles that are passively transported by only surface currents. As mentioned in the manuscript we do not incorporate additional transport processes due to e.g. wind, waves and tides or the 'inertial' effects that are due to the buoyancy, size or shape of the macroplastic. Including these processes with the aim to build a more realistic transition matrix currently still involves many parameterizations and further assumptions and even the application of the Maxey-Riley approximation to oceanographic scale is not yet fully worked out (e.g. Beron-Vera 2021). We do acknowledge, as stated in the manuscript, that these processes likely impact the resulting pathways and therefore the transition matrix, but chose not to include them because we choose to focus on the network-theory component.

Our reasoning for this is twofold. Firstly, it is not our aim to focus on the impact of different processes on the macroplastic pathways within the marine reserve as there are currently no

observational means available to validate or interpret the results. Second, the main aim of the paper is to introduce a network methodology that can assess the impact of cleanup strategies when a transition matrix is available. The methodology itself is therefore independent of changes in the transition matrix formation. As mentioned in the manuscript 'the method can be easily extended' by constructing a 'more accurate' transition matrix when including the relevant processes.

We do still specifically focus on macroplastic as removing larger items during beach cleanups is most effective and it allows us to restrict our analysis to surface currents and, based on the timescales of interest, neglect processes like biofouling and fragmentation.

> 2. *One weakness of the manuscript is the description of the methodology. Some of the definitions are not clear. For example the definition of retention rate, the loss rate, etc. I think the authors can greatly improve the definition of these centrality metrics through mathematical expressions, i.e using equations. For example using the mathematical formalism based on connectivity probabilities between network nodes in the weighted network, starting from the definition of the transition matrix.*

We agree that the retention rate, loss rate and beaching rate can be better understood when adding a mathematical definition. For consistency throughout the manuscript, we now use the edge weight definition to write the expressions for the retention rate ($p_{so=si}$), the loss rate ($p_{so,ocean}$) and the beaching rate ($1 - p_{so=si} - p_{so,ocean}$). We did not include mathematical expressions for the PageRank Centrality and the Betweenness Centrality as these are widely used and known centralities for which the interested reader is referred to the relevant literature.

> 3. *Have the authors considered that in temporal networks, as the analyzed here, also the synchronous arrivals at a node could impact on the network connectivity metrics? The "cumulated" implicit connectivity is based on adding up all the events of synchronous arrival (see Ser-Giacomi et al, 2021, PRE). However considering implicit connectivity could modify the resulting connectivity patterns*

We thank the reviewer for pointing us to this interesting work and have not yet considered investigating implicit connectivity. We feel that its application to the transition matrix based on connectivity of macroplastic transport between various coastlines and the corresponding interpretation is not straightforward and will likely result in a full study on its own. We have added a suggestion in the discussion section of the manuscript to further investigate this in the future.

> 4. *The resolution of the model is too coarse as to resolve submesoscale dynamics. Ignoring submesoscale motions is not a simple matter when it comes to surface material dispersion. It is well known that submesoscales are associated with vertical motion (an extreme case was documented via drifter measurements by D'Asaro et al., 2018 PNAS paper). The submesoscales cannot be removed from the analysis when their impact on the horizontal transport properties is substantial. They also generate high convergence zones, which could impact the connectivity probability between nodes. Please provide arguments to show that by dismissing small scale dynamics in the small region, the applicability of the results obtained here to the real ocean do not become very uncertain.*

The virtual particles are released using the surface currents of a 3D model simulation with a 4 km horizontal resolution. As described by Forryan et al. 2021, the model is able to resolve most of the relevant submesoscale variability and related frontal dynamics. The fact that we only use the surface currents for our studies does not mean that these are non-divergent, in fact, there are strong convergence and divergence zones displayed by the particle trajectories.

The non-resolved (< 4km) small scale dynamics is likely to only significantly impact the transport processes in the near-shore. This is also the region where additional coastal processes such as rip currents, swell, and wind shadow zones of islands become important, that are, regardless of the model resolution, not included. We fully acknowledge that by being able to resolve and include the effect of these processes for the transport of virtual macroplastic particles (which is still an active field of research) would make the transition matrix more accurate. However, as mentioned in our reply to comment (1), the aim is to introduce the network methodology once a transition matrix is formed, not to study the accuracy of the formation of the transition matrix itself.

The reviewer is right that the applicability of the presented 'most effective cleanup regions' for the Galapagos Islands, at this stage, is therefore limited. We agree that we can state this more clearly in the manuscript and in relation to this comment, comment (1) and (5) and comments from other reviewers, we have adjusted both the introduction, method and discussion section of the manuscript to make our aim of our work and applicability of our results clearer.

> 5. *Numerical diffusion. The spatial and temporal resolution of the model (4km and one day) could originate large numerical diffusion in the computation of the Lagrangian particle trajectories. Note that assuming velocities of 0.6m/s we obtain that in one day (the temporal resolution) the particle could move around 50km, which is 12 times larger than the spatial resolution (~4km). This can be "fixed" by decreasing the time step in the Lagrangian integration scheme, but still some small scale dynamics is missing, and this could affect to the Lagrangian transport associated with the large scale structures.*

In addition to the spatial resolution (comment (4)), our results are naturally also sensitive to the temporal resolution of the model simulations used for the particle tracking. We did already use a smaller time step for the Lagrangian integration scheme (1 hour) than mentioned by the reviewer and as we are using daily-mean surface velocity fields, the potential error in particle displacement is not as large as the reviewer suggests. Furthermore, previous studies have investigated the sensitivity of particle dispersion and connectivity to the temporal averaging of the ocean Lagrangian simulation and show that daily temporal resolution and daily particle releases are sufficient to incorporate connectivity fluctuations due to variable currents (e.g. Qin et al. 2014, Monroy et al. 2017).

Related to comment (4), we expect that mainly the transport in the near-shore is sensitive to the temporal resolution, as e.g. tidal effects become more important. We like to point the reviewer to the changes made in the discussion on the applicability of the presented results.

> 6. *One of the advantages of the methodology is that it can be used independently of whether there exists available macroplastic distribution data or not. However, a validation exercise could be beneficial, in particular to better support the conclusions.*

We fully agree with the reviewer here, but are unfortunately not able to validate our results at this stage as current available observational means are insufficient. There are still many challenges to overcome to be able to validate Lagrangian particle tracking simulations with observations of marine debris abundance at e.g. beaches or the ocean surface or surface drifter trajectories. As recently reported by Uhrin et al. (2022), challenges are for example the low spatial and temporal resolution of observations and the use of non-consistent measurements units.

Part of our future work will focus on the insights from a recent drifter field campaign to improve some of the parameterizations for macroplastic transport and beaching, but this is not the focus of the current manuscript. Furthermore, local authorities, in collaboration with the Galapagos Conservation Trust, plan to design a measuring campaign to quantify the non-local impact of

cleanups, which results will be vital to validate and can hopefully improve the presented methodology in the future.

> 7. *In the introduction section, I found some long and complicated sentences that could be split. Lines 30-32. Lines 37-39.*

Suggestion followed; both sentences are now split in the revised manuscript to improve readability.

> 8. *The sentence in line 40: "[...] the connectivity can still, to a first order [...]" could be improved. The connectivity by its self does not inform about aggregation but rather some derived metrics, and under some assumptions. Please clarify it.*

Suggestion followed, we've rephrased the sentence to: '*If the source distribution of macroplastic is unknown, the connectivity can still be used to derive which regions are likely to accumulate macroplastic and from which regions macroplastic is likely to spread to other coastlines, within the limits of the macroplastic flow model accuracy.*'

**References**

Beron-Vera, F. J. (2021). Nonlinear dynamics of inertial particles in the ocean: From drifters and floats to marine debris and Sargassum. *Nonlinear Dynamics*, *103*(1), 1–26. https://doi.org/10.1007/s11071-020-06053-z

Forryan, A., Naveira Garabato, A. C., Vic, C., Nurser, A. J. G., & Hearn, A. R. (2021). Galápagos upwelling driven by localized wind–front interactions. *Scientific Reports*, *11*(1), 1277. https://doi.org/10.1038/s41598-020-80609-2

Monroy, P., Rossi, V., Ser-Giacomi, E., López, C., & Hernández-García, E. (2017). Sensitivity and robustness of larval connectivity diagnostics obtained from Lagrangian Flow Networks. *ICES Journal of Marine Science*, *74*(6), 1763–1779. https://doi.org/10.1093/icesjms/fsw235

Qin, X., van Sebille, E., & Sen Gupta, A. (2014). Quantification of errors induced by temporal resolution on Lagrangian particles in an eddy-resolving model. *Ocean Modelling*, *76*, 20–30. https://doi.org/10.1016/j.ocemod.2014.02.002

Uhrin, A. V., Hong, S., Burgess, H. K., Lim, S., & Dettloff, K. (2022). Towards a North Pacific long-term monitoring program for ocean plastic pollution: A systematic review and recommendations for shorelines. *Environmental Pollution*, 119862. https://doi.org/10.1016/j.envpol.2022.119862

---

## Author Comment (AC4)

**Reply Reviewer #3**

*Not being a modeller myself, i have reviewed this article to the best of my abilities.*

*I think this paper is very well-written, clear and concise. The quality of the analysis, figures and data interpretation is very high and the scientific quality, rigour and significance seem to be excellent. I, therefore, congratulate the authors on such an excellent manuscript. Really, no issues were found during my review, and i think the approach and the results presented here, represent a very promising development in the field of plastic pollution. Besides, the real-world applications to the selection of the best clean-up locations are really concrete (although field validation of this modeling exercise is currently missing, but strongly envisaged).*

We thank the reviewer for going through our manuscript and the positive feedback.

*Maybe, i just have a minor comment for the author's consideration:*

*Given that as you say at lines 293-294 "locations with a high Retention Rate (Figure 7a), SSIsink (Figure 7d), betweenness centrality (Figure 7h) and PRin centrality (Figure 7i) are the most effective cleanup locations", why not combining all these 4 centralities in a single index to further restrict potential the list of potential clean-up locations to the "very" best ones? Assuming that clean-up capabilities of the Galapagos local municipality are restricted, this could be an effective way of further restricting/prioritising the best locations? (i.e. the ones having the best ranks in all 4 centralities). This index, could be potentially also added as a last panel to Fig. 7.*

One can indeed combine different centrality rankings to create an even higher impact as done for example in Pata et al. (2021). As suggested by the reviewer, we've combined the 4 most optimal centralities mentioned in our manuscript and compared the combined ranking to the individual rankings (see Fig. 1 in this document). For limited cleanup effort (<10% of coastline), this leads to a slightly higher benefit (Fig. 1a), but it needs more iterations to reach steady state as quickly as when using the individual Retention Rate centrality ranking (Fig. 1c). Combining different centrality rankings is not straightforward, as choosing *which* centrality rankings to combine and *how* depends on the impact objectives and available cleanup resources (e.g. the example of Fig.1 shows that the combined centrality ranking performs worse than some of the individual rankings when the cleanup effort > 10%). Furthermore, the intention of this paper was to introduce the network methodology for plastic pollution mitigation measures in a way that it could be reproduced in other regions. As it is not yet known whether the same centrality rankings score best when applied elsewhere, extending the analysis with combinations of the various centrality rankings is outside the scope of this paper, but worth mentioning as a recommendation for future work. The addition to the discussion can be found in the revised manuscript.

**References**
Pata, P. R., & Yñiguez, A. T. (2021). Spatial Planning Insights for Philippine Coral Reef Conservation Using Larval Connectivity Networks. *Frontiers in Marine Science*, *8*.
https://www.frontiersin.org/article/10.3389/fmars.2021.719691

[Figure]

*Figure 1: Adjustment from Fig.4 in the manuscript: A comparison of the impact metrics described in section 2.5 of the manuscript as a function of the fraction of coastline nodes cleaned for **all individual centrality rankings (grey lines)** and for **the combined ranking** of the Retention Rate-, SSIsink-, betweenness centrality-, and PRin-rankings **(red line)** . The benefit metric (a) measures the difference (in %) between the total number of particles removed and the number of particles removed if there would have been zero connectivity between the different nodes. The Left Behind on Land metric (b) indicates how many particles are still on land after steady state is reached. The Iterations metric (c) shows how many iterations where needed to reach steady state and provides an indication for how often one should clean.*